# Has the COVID-19 Pandemic Affected Cyberbullying and Cybervictimization Prevalence among Children and Adolescents? A Systematic Review

**DOI:** 10.3390/ijerph20105825

**Published:** 2023-05-15

**Authors:** Anna Sorrentino, Francesco Sulla, Margherita Santamato, Marco di Furia, Giusi Antonia Toto, Lucia Monacis

**Affiliations:** 1Department of Psychology, University of Campania, Luigi Vanvitelli, 81100 Caserta, Italy; 2Department of Humanities, University of Foggia, 71121 Foggia, Italy

**Keywords:** cyberbullying, cybervictimization, prevalence, COVID-19, children, adolescents, systematic review

## Abstract

In light of the alarming results emerging from some studies and reports on the significant increase in aggressive online behaviors among children and adolescents during the COVID-19 pandemic, the current research aimed at providing a more detailed evaluation of the investigations focusing on the cyberbullying prevalence rates published between 2020 and 2023. To this purpose, systematic searches were conducted on four databases (Web of Science, APA PsycInfo, Scopus and Google Scholar), and following PRISMA guidelines, 16 studies were included and qualitatively reviewed. Although studies were characterized by a large variety in cyberbullying operationalization and measurement, and by different methodologies used for data collection, the prevalence rates of the involvement in cyberbullying and/or cybervictimization generally revealed opposite trends: an increase in many Asian countries and Australia and a decrease in Western countries. The findings were also discussed by considering the effects of the COVID-19 pandemic. Finally, some suggestions were provided to policy makers for promoting prevention and intervention anti-cyberbullying programs in school contexts.

## 1. Introduction

Despite the absence of a universally agreed definition across researchers [1,2,3,4,5,6], cyberbullying—described as “an aggressive, intentional act carried out by a group or individual, using electronic forms of contact, repeatedly and over time against a victim who cannot easily defend him or herself” [7] (p. 376)—has emerged as a serious public health issue throughout the globe for children and adults alike. Being characterized by specific features, such as cyberbully anonymity, and the potential audience’s magnitude [8], the phenomenon has attracted international attention and concerns. Indeed, prior studies on its prevalence showed that more than a third of young people across 30 countries reported being a cybervictim [9], with rates ranging from 6.0 to 46.3% for cyberbullying and from 13.99 to 57.5% for cybervictimization [10]. Other investigations focused on several adverse behavioral, psychological and mental health outcomes [11,12,13,14,15,16,17,18], highlighting, for instance, the existence of a vicious circle that underlines how adolescents and young adults affected by pre-existing mental health problems (such as depression) are also more at risk of being cybervictimized [19] and, therefore, more inclined to exacerbate previous psychosocial problems [12,18]. Several studies also indicated significant associations between cybervictimization and/or cyberbullying involvement and internalizing and externalizing problems [15,20,21], depression [22,23,24], anxiety, abdominal pain, loneliness, lower self-esteem, life satisfaction, hyperactivity [11,15,25,26,27], stress [28,29], post-traumatic stress symptoms [30], poor academic performance [31,32], suicidal ideation and suicide [33,34,35,36]. Concerning the negative behavioral consequences, youth involvement in both cyberbullying and cybervictimization is associated with juvenile delinquency (sexual frequency, alcohol consumption, marijuana usage and cannabis smoking) [37,38,39] and violent/deviant or aggressive behaviors toward others [40,41,42,43,44].

The picture of such a phenomenon becomes sadder when considering the contextual factor of the COVID-19 pandemic that negatively affected children’s psychological health due to the prolonged school closures and the consequent social isolation [45]. Indeed, adolescents and young people’s increased use of the Internet and information and communication technologies (ICT) during the COVID-19 pandemic could have posed a greater risk of being involved in cyberbullying and cybervictimization behaviors, thus exacerbating the likelihood of displaying mental health problems. This is in line with the previous empirical evidence showing that spending much time online and engaging in online activities are crucial risk factors for cyberbullying and cybervictimization [15,16,46,47,48,49,50].

The hypothesized close connection between time spent on digital devices during the pandemic and the increase in cyberbullying has led some researchers to provide empirical evidence [51,52,53]. In this vein, inspired by the general strain theory (GST), that posits strains or stressors increase negative affect due to exposure to negative stimuli and consequently predict broader aggressive behaviors, Barlett et al. [51] examined the mediating role of stress in the relationship between COVID-19 experiences and antisocial behavior (cyber-aggression). The results provided tentative support for the theory among the US adult population by collecting data on public online discussions. Han et al. [52] showed a general increase in cyberbullying prevalence among Chinese children and adolescents, indicating that about 11.0% reported being cybervictimized, whereas 5.0% of participants said they had cyberbullied others during the COVID-19 outbreak in 2020. 

A similar and considerable increase was also documented by Karmakar et al. [53], who assessed the impact of COVID-19 on the reported cyberbullying incidents through user-gendered tweets. In the same data-driven direction, the study by L1GHT [54], focused on the analysis of the increase in online toxicity as a result of the pandemic, reported a 900% increase in hate speech on Twitter toward China/Chinese people, a 70.0% increase in cyberbullying among children in chat forums and a 40.0% increase in toxicity on gaming platforms.

Moreover, a European technical report [55] aimed at providing a snapshot of how children across 11 countries experienced different online risks during the COVID-19 lockdown by comparing changes that occurred before and during the crisis underlined that almost 49.0% of the children declared having been cyberbullied at some point and that, at the national level, the highest percentages were recorded in Italy and Ireland (59%) followed by Germany (58.0%) and other countries. A further interesting note is that by comparing the two periods (pre- and during the pandemic), nearly half (44.0%) of children who had already been cybervictimized declared an increase in the phenomenon during the first wave of the pandemic. In contrast, more than a fifth of the children (22.0%) perceived the phenomenon as lesser than before. These results seem to reveal an evident contradiction among European children in estimating their experience with upsetting situations online. Conversely, the UNICEF Canadian report [56] indicated an opposite trend of its prevalence, reporting a 17.0% reduction in cyberbullying during the pandemic. 

Given the social alarm raised by some empirical evidence and, to the best of the authors’ knowledge, the lack of a detailed evaluation of cyberbullying prevalence rates across studies carried out during the COVID-19 pandemic, the present work aims to systematically review studies that investigated the prevalence of this phenomenon. In fact, restrictions due to the COVID-19 pandemic have limited children and adolescents’ face-to-face interactions, potentially increasing the chances of being involved in cyberbullying and cybervictimization rather than in school bullying. Although school bullying prevalence rates were, for a long time, much higher than cyberbullying [57,58], the lockdown restriction measures made engagement in school bullying “unfeasible” [59,60].

Understanding cyberbullying and cybervictimization trends during the COVID-19 pandemic would be crucial to designing and implementing primary and secondary prevention activities.

## 2. Methods

The systematic review included studies published from 2020 to 2023 (also considering any in-press publications on the topic). The Preferred Reporting Items for Systematic Reviews and Meta Analyses (PRISMA) guidelines [61] were adopted for searching, assessing study eligibility and reporting the results.

### 2.1. Search Strategy and Selection Criteria

A systematic search was conducted by four researchers in December 2022 on four databases: Web of Science, APA PsycInfo, Scopus and Google Scholar. To extract significant data for our research purposes, specific keywords were included in the search string to be launched on the selected databases: “cyberbullying”, “cybervictim*“, “prevalence”, “diffusion”, “minor*“, “adolescent*”, “teenager*”, “COVID-19”, “SARS-CoV-2”, “pandemic” and “lockdown”. These keywords were combined utilizing Boolean operators to create a search string. The search string used in all four research databases was (cyberbully* OR cybervictim*) AND (prevalence OR diffusion) AND (minor* OR adolescent* OR teenager*) AND (“COVID-19” OR “SARS-CoV-2” OR pandemic OR lock-down). Researchers (M.S., M.d.F.) individually checked all the titles and abstracts collected from the databases. They afterward discussed limits, inconsistencies and standards until a consensus was reached.

Articles identified were screened as abstracts. After excluding those that did not meet our inclusion criteria, the full texts of the remaining articles were assessed for eligibility, and decisions were made regarding their final inclusion in the review. 

The inclusion criteria were as follows: (1) studies written in English and peer-reviewed; (2) studies published between 2020 and 2023 (including any publications in press); (3) focusing on a survey population aged 6–18 years; (4) aimed at assessing the incidence of cyberbullying and/or cybervictimization among school-aged students during the pandemic. 

On the other hand, exclusion criteria for this study are as follows: (i) reviews, case studies and case series; (ii) articles not published in English; (iii) focusing on general features of cyberbullying not during COVID-19 pandemic; (iv) studies focusing on cyberbullying and/or cybervictimization against minorities (i.e., ethnic/racial, LGBTQ+, SEN, disability, etc.). 

### 2.2. Data Extraction 

Two independent researchers (M.d.F. and M.S.) separately carried out data extraction, reading full-text articles to include only relevant objects. In case of disagreement, a third researcher (either A.S. or F.S.) was consulted to reach a consensus. Literature was only rejected and excluded if at least two researchers agreed it was not relevant.

To manage the records, Rayyan review-management software was utilized [62]. The variables extracted included lead author/year, country, study design (cross-sectional and longitudinal), sample size, sex (% males), age (mean ± SD, range), period of data collection (presence or absence of lockdown measures), instruments, quality appraisal (see below) and key findings.

### 2.3. Quality Appraisal 

To check the quality appraisal, the Critical Appraisal tools in JBI Systematic Reviews for prevalence studies were used [63] (Appendix A). The tool aims to assess a study’s methodological quality and determine the extent to which a study has addressed the possibility of bias in its design, implementation and analyses by evaluating 9 domains: appropriateness of the sample frame, participants’ recruitment, simple size adequateness, participants and setting described, sample coverage of data analyses conducted, method validity, suitability of statistical analyses, response rate and missing management. Two researchers (A.S. and F.S.) independently appraised the 16 studies included.

All the studies assessed were considered to be eligible as a result of the overall appraisal.

## 3. Results

The initial search identified a total of 4191 records. Before the screening phase, a total of 4129 records were removed since 4123 were remarked as ineligible, and 6 were duplicates. Thus, a total of 62 papers were scrutinized based on the search criteria, title and abstract. Twenty reports were excluded. Therefore, the remaining 42 full texts were considered potentially eligible for the study. 

Twenty-six articles were excluded due to not meeting the inclusion criteria: two studies focused on wrong outcomes [64,65], ten were conducted before the COVID-19 pandemic [66,67,68,69,70,71,72,73,74,75], seven articles involved participants aged > 18 years [51,76,77,78,79,80,81], two articles included participants aged > 18 years and examined the phenomenon before the COVID-19 pandemic [82,83], three did not report the period of data collection nor made any reference to the COVID-19 pandemic [84,85,86], and two studies were no empirical investigations, being a scoping review [87] and a systematic review [88] (Appendix A). 

The results of the search are shown in a PRISMA 2020 flow diagram (Figure 1). 

### 3.1. Geographical Distribution, Period and Methodologies of Data Collection

As shown in Table 1, among the articles included in the present systematic review, ten were conducted in Asia, that is, three in China [52,89,90], two in South Korea [91,92] and the remaining in Indonesia [93], Malaysia [94], Thailand [95], Turkey (eastern Anatolia) [96] and Vietnam [97]; three studies were conducted in European countries, i.e., Croatia [98], Finland [60] and Germany [99]; and the remaining three studies were carried out, respectively, in Australia [100], Canada [101] and Chile [102]. 

Regarding the selected studies’ research design, four studies were longitudinal investigations [60,91,92,100], one was a randomized retrospective study [101], and the others were cross-sectional [52,89,90,93,94,95,96,97,98,99,102].

The number of school-aged students involved ranged from 107 in Germany [99] to 34.771 in Finland [60]. Fifteen studies involved middle and/or high school students with four also including primary school pupils [52,60,92,101]; one study involved only elementary school students [91].

The time ranges of the data collection varied across the studies reflecting the spread of COVID-19 that followed different rates for each geographic area, based on random and epidemiological factors, as underlined by the Johns Hopkins University research team (https://coronavirus.jhu.edu/data/animated-world-map, accessed on 10 May 2023). Two studies [94,96] reported that data collection took place in 2021, five studies [60,95,97,99,100] were conducted during the early lockdown phase of 2020 and/or shortly after the pandemic outbreak in the spring–summer 2020 (March–August), and six studies were conducted between autumn 2020 and winter 2021 [89,90,92,93,101,102]. Across the two Chinese studies, the data collection period did not overlap, ranging from September–October 2020 [90] to November 2020–January 2021 [89]. In the longitudinal study involving South Korean elementary school students, the authors referred to the first data collection (baseline) in 2019 and a follow-up in 2020 [91]. 

Finally, two investigations [52,98] did not clearly report the timeframe even if explicit references to the pandemic were present in the title; thus, such information was inferred by consulting indirect sources or textual and/or metadata elements. Indeed, for the former [52], involving participants from the rural region of Shandong in China, since the article was submitted to the journal on February 5th, 2021, we can hypothesize that data collection occurred in 2020. In addition, considering that the questionnaire administered to participants included the item “Have your classmates or peers implemented these actions to you since January 2020?”, we can infer that the data collection took place during or just after the Shandong “strictest lockdown” phase, which occurred roughly from late January 2020 to early March 2021, as indicated in other similar Chinese studies [103,104]. For the latter study [98], although the range period was not explicitly indicated, it was reported that pupils were invited to complete an online survey sent to 15 schools in December 2020.

Concerning the data collection procedure, most studies utilized online surveys, and three [89,96,97] collected data during school hours.

In particular, Eroglu et al. [96] reported that data were collected by psychological counselors during school hours, even if the rate of students attending school was low because they worried about both contracting COVID-19 and transmitting it to their families. Xiang et al. [89] collected data by administering a paper-and-pencil questionnaire to Chinese adolescents during school hours. Thumronglaohapun et al. [95] sent envelopes via mail to each institute participating in the survey to let the students decide when and where to fill in the questionnaires. In the study conducted by Thai et al. [97], data were collected while schools were open, but the procedure was not specified. 

In the study carried out in Australia [100], data on CB/CV prevalence were collected adopting the Cyberbullying Questionnaire–Revised (CBQ-R) [40] from 2015 to 2019, while the Cyber Bullying Participant Roles Scale [105] was administered online in 2020. Moreover, Shin and Choi [92] and Choi, Shin and Lee [91] assessed CB/CV in person through teachers in 2019, while in 2020, data were collected using an online version of the same questionnaire because of the COVID-19 pandemic. 

Another important methodological issue is related to the definitions adopted to operationalize CB and CV, which, as summarized in Table 1, differed across the selected studies. Such definitions were analyzed referring to Chun et al.’s categorization [2]: (1) the use of electronic means or devices, (2) vulnerability (e.g., those who cannot easily defend themselves), (3) repeated harm or behavior, (4) deliberate or intentional act, (5) unwanted information of others and (6) the purpose of threatening/harassing/embarrassing others. Thirteen studies included the use of electronic means or devices in the definition for CB and/or CV [52,60,89,90,91,92,94,96,97,98,99,100,101]; five definitions referred to victims’ vulnerability and incapacity to defend themselves [52,90,94,95,100]; six studies defined it as repeated harm caused to victims [52,89,90,98,99,100]; six studies also included the perpetrator’s deliberate intention to harm in the definition [89,94,96,98,100,101]; and five definitions referred to cyberbullies’ purpose of threatening/harassing/embarrassing others [89,90,91,92,97]. None of the sixteen studies used “unwanted information of others” in their CB definition.

Only five studies [60,90,91,94,100] provided a definition of CB to students along with the self-report questionnaires utilized to assess their involvement. In contrast, the remaining 11 studies did not explicitly report if a description of CB/CV was provided before data collection. However, multi-item scales were used to overcome such limits by asking participants to rate the frequency of specific CB/CV behaviors [106,107]. 

Two studies [93,94] did not explicitly report if a CB/CV definition was provided to participants before collecting data and adopted a single item to measure students’ engagement in CB and/or CV.

### 3.2. Data on Prevalence of Cyberbullying/Victimization during the Pandemic

At first glance, almost half of the studies [60,89,90,93,96,100,102] did not report CB and/or CV prevalence rates, and the presented data varied considerably in terms of CB and/or CV percentages (Table 1). In addition, a comparison among them turned out to be difficult, since different methods were used to operationalize and measure such constructs. 

Indeed, concerning the psychometric tools assessing CB/CV, more than two-thirds of the studies used validated measures of cyberbullying and cybervictimization, three studies developed ad hoc questionnaires [52,91,92], and one study adopted a single item for involvement in cyberbullying, cybervictimization and cyberbullying/victimization [93], whereas in another one [60], two single items were adopted for measuring cybervictimization only. 

Such heterogeneous measurement also implied that not all studies reported prevalence rates concerning all four possible categories of involvement in the phenomenon (i.e., not involved; cyberbully; cybervictim; cyberbully/victim). As shown in Table 2, only two studies reported participants’ involvement in cyberbullying, referring to the four involvement categories [52,98].

Three studies reported percentages of participants not involved in any cyberbullying episodes during the pandemic, with the highest rate found in China [52] (86.68%), followed by the Croatian study (73.08%) [98], while the lowest percentage was found among Thai students (50.8%) [95].

Seven studies reported the percentage of cyberbullying. The two longitudinal studies conducted in South Korea found the highest rates, reporting that 12.4% of elementary [92] and 9.5% of school-aged students [91] were involved in cyberbullying during the pandemic. The lowest CB involvement percentage rate was found in China [52], with 1.89% of 1111 participants declaring being cyberbullies.

Eight studies reported data concerning cybervictimization diffusion. The highest percentage was found in Thailand (49.2%) [95], followed by Vietnamese students (36.5%) [98], while the lowest rate was reported in the Chinese study by Han et al. [52] (8.19%), followed by the percentage found in Croatia (12.75%) [98].

Five studies reported the percentage of participants’ involvement as cyberbullies/victims [52,91,92,94,98]. The highest percentage of cyberbullies/victims (9.7%) was reported by the longitudinal study involving South Korean elementary school students [91], followed by the one that was conducted in Croatia (8.3%) [98], with Malaysian students showing the lowest involvement as cyberbullies/victims (2.4%) [94].

Furthermore, systematic comparisons between CB and/or CV prevalence rates before and after the pandemic across these 11 cross-sectional studies are not possible. Therefore, the emerging results are compared with those observed in previous investigations if they are explicitly discussed and reported.

The investigation carried out by Thai et al. [97] in May 2020, when all schools were open, found that (i) cybervictimization was identified in 36.5% of 1492 Vietnamese adolescents and almost one-fourth of students experienced multiple types of cybervictimization; (ii) cybervictims had 1.81 times (95% CI [1.42–2.30]) higher odds of reporting symptoms of depression; and (iii) a higher likelihood of having depression was also found among female students who had Internet addiction. By comparing findings with others collected before the pandemic period, it emerged that the prevalence rate resulted higher than those collected in Vietnam, i.e., in Hue [108] (9.0%; of 648) and in northern provinces (24.0% of 763 adolescents) [109]. The higher prevalence rate was attributed to the higher level of depression found among adolescents, whereas the lower rate was attributed to the different Internet development and usage patterns.

In the Chinese context, 13.32% of 1111 adolescents reported being involved in the phenomenon, with 8.19% being involved in cybervictimization, 1.89% being involved in cyberbullying and 3.24% being cyberbullies/victims [52]. The reported prevalence rates were generally higher compared to prior studies carried out in China, highlighting a victimization prevalence rate ranging from 5.51% [110] to 7.49% and a perpetration rate of 2.05% [13]. According to the researchers, the higher prevalence rates in these phenomena were due to intensive use of the Internet service under the age of 10 and greater loneliness due to social distancing.

A similar conclusion was drawn in the Malaysian study involving 1290 high school students [94]. The researchers found that 3.8% of the participants were cyberbullies and 13.7% were cybervictims. Prevalence rates were lower compared to previous investigations conducted in the same country. In particular, a study carried out in the state of Negeri Sembilan found a prevalence rate of 52.2% in cybervictimization [111], while a study conducted in the state of Penang reported a prevalence rate of 20.9% in cyberbullying and 31.6% in cybervictimization [112]. According to the authors [94], cyberbullying prevalence varies across the states as some states, particularly the more urban ones, may have greater access to technology, thus leading to a higher prevalence of cyberbullying. Moreover, in this investigation [94], a higher rate of suicide prevalence was found compared to previous studies conducted in the same country, highlighting that adolescents who had been cybervictimized showed an increased risk of depression and anxiety symptoms and were more prone to engage in suicidal behaviors than those who had not been cybervictimized.

In the study carried out in Germany [99], 107 adolescents reported their cybervictimization frequency, with 25.7% indicating experiencing cybervictimization less frequently, 54.3% indicating experiencing cybervictimization equally frequently and 20.0% indicating more frequent cybervictimization during the pandemic as compared to before. It should be noted that the percentage indicating less cybervictimization was inferred from participants’ self-report and might be an underestimation of the true decrease in cybervictimization. In the same study, greater levels of cybervictimization were found to be related to lower emotional self-efficacy, and lower self-efficacy was related to lower self-esteem. Moreover, cybervictimization was related to lower well-being in this sample.

Similarly, in a retrospective randomized study [101], the results showed a significant decrease in CB and CV involvement among Canadian participants during the COVID-19 pandemic compared to the pre-COVID-19 situation: indeed, with regard to the cyberbullying involvement, 13.8% of participants reported to be involved as perpetrators before the pandemic compared to the 2.3% during the pandemic. The same reduction emerged for cybervictimization (13.8% vs. 11.5%). These results are consistent with the recent Canadian UNICEF report [56].

### 3.3. Trends in the Prevalence of Cyberbullying/Cybervictimization before and during the Pandemic

To track the trends in the prevalence of cyberbullying/cybervictimization before and during the pandemic, four longitudinal investigations are scrutinized, although the diversity in sample size and methodological and measurement strategies make it difficult to compare the observed data. Indeed, the study carried out by Repo et al. [60] assessed the prevalence rate of cybervictimization only, the two Korean studies [91,92] were part of a broader national project on cyberbullying and cybervictimization started in 2019, and the last one reported CB and CV prevalence at four time points from 2015 to 2020 [100]. Concerning the first investigation [60] carried out on a sample of 34,771 students attending the Kiva anti-bullying program, the findings showed a decrease in cybervictimization with the average rate decreasing from 2.0% before the pandemic to 1.0% during the lockdown. An opposite trend was reported in the two Korean studies [91,92], where the rates of the prevalence of cybervictimization increased from 19.0% in 2019 to 19.7% in 2020 [92] with the highest prevalence increasing from 25.8% in 2019 to 32.7% in 2020 among elementary school students [91]. Conversely, a general decrease was recorded in cyberbullying when looking at different age groups from 18.0% in 2019 to 9.5% in 2020 and in cyberbullying/victimization involvement from 10.1% in 2019 to 6.4% in 2020 [92]. These results seem to confirm the negative impact of COVID-19 of increasing cybervictimization diffusion among primary school pupils.

The study carried out in Australia from 2015 to 2020 [100] showed an increase in the trend of cybervictimization over time, with higher rates in 2017 and 2019 compared to 2015; CV rates in 2020 were also higher than those reported in 2015 and 2019. By contrast, a different pattern emerged in the prevalence rates of cyberbullying, highlighting a significant increase only in 2020 compared to the timeframe 2015–2019.

To sum up, three studies found an increasing trend with two studies reporting a significant increase in cybervictimization behaviors [91,92] and one study reporting an increase in the diffusion of both cyberbullying and cybervictimization [100]. On the other hand, only one study showed a decreasing trend in the involvement in cybervictimization among children and youth [60]. An overview of all studies is provided in Table 2.

### 3.4. Gender Differences in Cyberbullying and Cybervictimization Involvement

Half of the analyzed studies indicated gender differences in the involvement in cyberbullying and cybervictimization during the COVID-19 pandemic, although mixed results emerged: three studies found that males were more likely to be involved in cyberbullying [89] both before and during the pandemic [92,100]. A different pattern was found in the study carried out by Choi et al. [91] involving only elementary school pupils, indicating that girls were more likely to be involved in cyberbullying before and during the COVID-19 pandemic. Two studies reported no significant gender differences in cyberbullying involvement [98,101]. 

Concerning cybervictimization, three studies found no significant gender differences [99,101,103], and one study found a significant association between being male and cybervictimization [89], whereas Vaillancourt et al. [101] found that, both before and during the COVID-19 pandemic, girls scored higher in cybervictimization.

## 4. Discussion 

The COVID-19 pandemic has brought about a rapid and drastic change in human relationships, especially among children and young people, due to the prolonged closure of schools and the long-lasting social isolation. In this regard, several studies highlighted the negative impact of COVID-19 on children and youth mental health and an increase in youth psychological distress, worry, loneliness, anxiety, depression and traumatic symptoms [45].

It is conceivable that these adverse psychological outcomes could have been affected or worsened by the youth’s increased involvement in cyberbullying and cybervictimization, as underlined by documents released in the initial stages of the pandemic [55,113]. Indeed, especially during lockdown periods, children and adolescents increased their screen time, thus increasing their risk of being involved in cyberbullying and/or cybervictimization in more severe ways [10,114]. Furthermore, the significant overlap and inter-correlation between cyberbullies and cybervictims’ roles [43] potentially create a recursive negative circle that may cause several negative and deviant behaviors that could in turn result in internalizing and externalizing symptoms [11].

Given the lack of a review concerning the incidence of this phenomenon during the pandemic period, the current qualitative synthesis attempted to fill this gap by analyzing the prevalence rates of cyberbullying and/or cybervictimization among children and adolescents across countries, bearing in mind the different variations of the SARS-CoV-2 transmission in each geographic area. Sixteen studies resulted eligible for our investigation. Ten studies were conducted in Asian countries, three in Europe and the remaining three in Australia, Canada and Chile. Although the selected studies could provide a global overview for a deeper understanding of the prevalence rates of the phenomenon, the large variety of the operationalization and measurement of the construct and different methodologies used for data collection limited the evaluation, thus adding further complexity to the phenomenon. 

Indeed, despite prior recommendations suggesting the need to follow shared criteria or categorization in defining cyberbullying before collecting data [2,106,107,115], from the current qualitative synthesis, it emerges that only five studies were in line with this methodological orientation.

Concerning the prevalence rates, the review generally indicated two opposite trends: an increase, for example, among Australian teenagers in both CB and CV [100], and a decrease among South Korean primary schoolers in CV [91] in the context of a general decrease in CB/CV among South Korean elementary, middle and high school students [92]. A decreasing trend was also reported among Canadian children and adolescents in both CB and CV [101] and among Finnish adolescents in CV involvement [60].

These findings might reflect the cross-cultural differences as well as the different impact and severity of the lockdown measures during the pandemic. Studies underlining a decrease in CB and/or CV during the pandemic seem to corroborate the arguments proposed by Olweus and Limber [106], who rejected the general claim “Screen time is up - and so is cyberbullying” [113] thus stressing that cyberbullying involvement is not a mere consequence of the increased time on digital devices but a long-established problem. Moreover, such a reduction in CB and CV involvement might reflect the fact that, as a consequence of the lockdown measures, parents and teachers were more involved in closely monitoring young people’s virtual activities [100,116,117]. By contrast, studies that found an increasing trend in CB and CV prevalence rates hypothesized the contributing role of factors such as higher levels of loneliness due to social distancing, increased time spent online and poor levels of psychological well-being [52,94,97,99]. Such findings could foster the general assumption that cyberbullying is compounded by the interplay of different individual, relational and contextual factors [118]. In this vein, investigations carried out by Shin and Choi [92] and Choi, Shin and Lee [91] highlighted that the COVID-19 pandemic exposed especially primary school pupils to a greater risk of CB and CV, shedding light on the importance of including children in anti-cyberbullying prevention and intervention programs. The development and implementation of such programs with this population could help to pinpoint effective primary prevention strategies that may foster an increase in protective factors toward future youth and adolescents’ involvement in CB and/or CV. In this direction, primary schools could represent the place of choice for the implementation of effective primary prevention actions, able to inform and sensitize children to the “safe” use of electronic devices and social media and to avoid the psychosocial consequences (e.g., depression, anxiety) resulting from the involvement in cyberbullying and/or cybervictimization [119]. In this direction, Repo et al.’s [60] results remarked the significant role played by the KiVa anti-bullying program [120] in promoting greater awareness of risks related to online settings and, therefore, in lowering cybervictimization rates. 

## 5. Conclusions

Considering the importance of developing and delivering anti-cyberbullying programs, primary prevention and awareness-raising activities about online risks for children and adolescents need to be addressed at governmental and community levels. In this regard, considering the increasing use of social media and the Internet by younger users or “digital natives”, particular attention should be paid by social media platforms in terms of defining and setting clear anti-cyberbullying policies. From an educational perspective, social media allow users to easily connect with knowledge, share it and transform and rework content—internalizing it and making the learning flow smoother [121]. However, even if some popular social media platforms set restrictions for young children, their use can still become potentially dangerous for children’s psychological and physical well-being. Indeed, a study based on a qualitative analysis of 14 social media companies’ policies and interviews with social media company representatives found that cyberbullying continues to take place regardless of social media policies [122]. Although some social media platforms have improved their policies over time, by implementing methods to report cyberbullying behaviors and encouraging non-anonymous experiences by making users use their real names [123], specific curricula focusing on media education and awareness of online risks should be included in cyberbullying prevention and intervention programs [124].

Furthermore, future research should therefore focus on developing, implementing and assessing the long-term efficacy and sustainability of multi-componential CB and CV programs targeting (also) primary school students. 

Although the aim of the current systematic review was not fully achieved due to the above-mentioned limitations arising from the qualitative evaluation of the selected studies, it could contribute to providing a partial snapshot of the prevalence rates of involvement in cyberbullying and/or cybervictimization across the countries during the pandemic period and to remarking the need for a shared agreement on how to define and measure CB and CV [2]. Furthermore, our results stress the need for future research assessing retrospectively the possible long-term impact of the pandemic period on CB and CV prevalence. 

## Figures and Tables

**Figure 1 ijerph-20-05825-f001:**
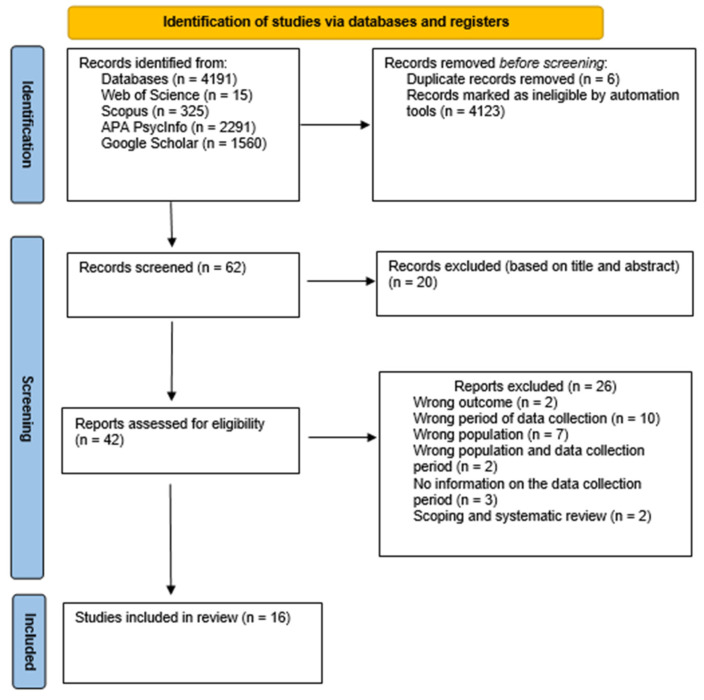
PRISMA 2020 flow diagram.

**Table 1 ijerph-20-05825-t001:** Definitions and measures of CB/CV.

Study	Period of Data Collection	Procedure	Definition of CB/CV	Measure	Item CB	Item CV	Item CB/CV
Choi et al. [91] South Korea	2020	Online survey	Act of bullying or persistent harassment on the Internet, as well as stalking, defamation and other forms of online harassment	Ad hoc questionnaire	6 CB behaviors assessed through dichotomous answers: “Experienced” and “Not experienced”	6 CV behaviors assessed through dichotomous answers: “Experienced” and “Not experienced”	-
Eroglu et al. [96] Turkey	November 2021	School-based survey	Cybervictimization is the intentional use of negative online behaviors against a person as a result of teenage hostility and online collaboration	The Revised Cyberbullying Inventory	-	14 items on a 4-point Likert scale (from “never” to “more than three times”)	-
Mohd Fadhli et al. [94] Malaysia	May–September 2021	Online survey	An offensive, planned act committed by a group or individual using electronic communication technology against a victim who is helpless to protect themselves on a frequent and ongoing basis	The Malay version of the Cyberbullying Scale	One single question asking “Have other children used any of the following items to bully you?” Students rated their involvement in CB by answering 14 items scored on a 5-point Likert scale (from “0 = never” to “4 = always”)	One single question asking “Have you used any of the following items to bully other children?” Students rated their involvement in CV by answering 14 items on a 5-point Likert scale (from “0 = never” to “4 = always”)	-
Repo et al. [60] Finland	Spring 2020	Online survey	Over time, bullying that occurs offline often spreads to online settings	Two ad hoc items	-	“Did your schoolmates bully you during remote schooling?” “Have your peers from school bullied you during this year, before the remote schooling began?”	-
Rodriguez-Rivas et al. [102] Chile	October–November 2020	Online survey	-	Ybarra et al.’s scale	-	A 4-item self-report scale (from “1 = not sure” to “5 = Often”)	
Schunk et al. [99] Study 1 Germany	May–June 2020	Online survey	Repeated and intentional pain caused by using computers, cell phones and other electronic gadgets	The Mobbing Questionnaire for Students	-	4 items on a 5-point Likert scale (from “1 = never” to “5 = few times a week”) For each item, participants rated frequency of CV by comparing their experiences before and during COVID-19	-
Thai et al. [97] Vietnam	May 2020	School-based survey	Any form of harassment that causes people emotional anguish through emails, chat rooms, websites or messaging is referred to as cyberbullying	Cyber Bullying Scale	-	16 items on a 5-point Likert scale	
Thumronglaohapun et al. [95] Thailand	May–August 2020	School and/or online	Cyberbullying is misuse that can be detrimental to the cyberbullied individuals’ mental health and lifestyle, and it often ends up with the victim becoming depressed and fearful of society and, in the worst cases, experiencing suicidal ideation	European Cyberbullying Intervention Project Questionnaire	12 items assessing the following CB frequency ranges: 0, 1–5, 6–10, 11–15 and >15 times/year	-	-
Trompeter et al. [100] Australia	May–August 2020	Online survey	Cyberbullying is a repetitive, premeditated, aggressive conduct carried out via technologies that make it difficult to defend oneself	Cyber Bullying Participant Roles Scale	9 items on a 6-point scale (from “1 = not at all” to “6 = many times a week”)	9 items on a 6-point scale (from “1 = not at all” to “6 = many times a week”)	-
Wiguna et al. [93] Indonesia	August–October 2020	Online survey	-	3-item scale adapted from other questionnaires	One question asking “During the past 6 months, how often have you been cyber-bullied?” Students answered using a 4-point Likert scale (from “1 = never” to “4 = almost daily”)	One question asking “During the past 6 months, how often have you cyber-bullied others?” Students answered using a 4-point Likert scale (from “1 = never” to “4 = almost daily”)	One question asking “During the past 6 months, how often have you been cyber-bullied and being cyber-bullied others?” Students answered using a 4-point Likert scale (from “1 = never” to “4 = almost daily”)
Xiang et al. [89] China	November 2020–January 2021	School-based survey	Using information and communication technologies to repeatedly and intentionally harm, harass, hurt and/or embarrass a target	Chinese version of the E-Bullying and E-Victimization Scale (E-BVS)	6 items on a 7-point Likert scale (from “0 = never” to “6 = 6 times or more”)	6 items on a 7-point Likert scale (from “0 = never” to “6 = 6 times or more”)	-
Zhao et al. [90] China	September–October 2020	Online survey	Cyberbullying can be said to be an extension of traditional bullying through online platforms, which refers to individuals or groups communicating by sending electronic messages or other ways, in order to attack and harm vulnerable groups repeatedly who cannot protect themselves on the Internet	Cyberbullying Scale	12 items measuring direct and indirect CB on a 5-point Likert scale (from “1 = never” to “5 = always”)	-	-
Han et al. [52] China	Not explicitly reported	Online survey	Cyberbullying refers to activities committed in cyberspace or utilizing information communication technology with the intent of causing harm to those who are unable to defend themselves	Ad hoc questionnaire	6 dichotomous items (0 = “never” and 1 = “rarely”, “sometimes” and “often”)	6 dichotomous items (0 = “never” and 1 = “rarely”, “sometimes” and “often”)	-
Shin and Choi [92] South Korea	October–November 2020	Online survey	Any action that causes distress to another person in the cyber environment (such as Internet, cellular phone) via language and video, among other aspects	Ad hoc questionnaire	7 CB behaviors assessed through dichotomous answers (Experienced/Not experienced)	7 CV behaviors assessed through dichotomous answers (Experienced/Not experienced)	-
Vaillancourt et al. [101] Canada	September–November 2020	Online survey	Bullying can be physical, verbal, social or online. A student who bullies wants to hurt the other person, and they do it more than once and in an unfair way. Sometimes, a group of students will bully another student. It is not bullying when two students of about the same strength or popularity have an argument or disagreement	Adapted version of the Olweus Bully/Victim Questionnaire	5 self-report items measured on a 5-point scale (0 = not at all to 4 = many times a week)	5 self-report items measured on a 5-point scale (0 = not at all to 4 = many times a week)	-
Vejmelka and Matković [98] Croatia	December 2020	Online survey	Cyberbullying as repeated and intentional harm to people through a computer, cell phone or other electronic device	European Cyberbullying Intervention Project Questionnaire	11 items on a 5-point Likert scale (from “0 = never” to “4 = more than once a week”)	11 items on a 5-point Likert scale (from “0 = never” to “4 = more than once a week”)	

Notes: CB = Cyberbullying; CV = Cybervictimization.

**Table 2 ijerph-20-05825-t002:** Overview of the selected 16 studies.

	Method				%			
Study	N (% of M)	Age Range	School Level	Design	NI	CB	CV	CB/CV
Choi et al. [91] South Korea	N = 1.738 (50.4%)	10–12 years	Elementary school students	Longitudinal study		12.4%	25.8%	9.7%
Eroglu et al. [96] Turkey	N = 337 (50.1%)	14–19 years (M = 16.56)	High school students	Cross-sectional study	-	-	-	-
Mohd Fadhli et al. [94] Malaysia	N = 1.290 (29.8%)	13–17 years (M = 14.48)	High school students	Cross-sectional study	-	3.8%	13.7%	2.4%
Repo et al. [60] Finland	N = 34.771	10–16 years	Primary and middle school students	Longitudinal study	-	-	-	-
Rodriguez-Rivas et al. [102] Chile	N = 287 (60.3%)	14–18 years (M = 15.95, SD = 1.13)	Middle school students	Cross-sectional study				
Schunk et al. [99] Study 1 Germany	N = 107 (35.5%)	13–18 years (M = 15.76)	Secondary school	Cross-sectional study	-	-	32.7%	-
Thai et al. [97] Vietnam	N = 1.492 (44.7%)	-	Secondary (71.8%) and high (28.2%) school students	Cross-sectional study	-	-	36.5%	-
Thumronglaohapun et al. [95] Thailand	N = 2.683 (30.5%)	M_age_ = 16.0	High school students	Cross-sectional study	50.8%	-	49.2%	-
Trompeter et al. [100] Australia	N = 159 (74.0%)	11–16 years (M = 13.0)	-	Longitudinal study	-	-	-	-
Wiguna et al. [93] Indonesia	N = 464 (33.3%)	11–17 years (M = 14.61, SD = 1.65)	Primary (3.2%), middle (58.0%) and high (38.8%) school students	Cross-sectional study				
Xiang et al. [89] China	N = 425 (57.9%)	11–18 (M = 15.06, SD = 1.48)	Middle (27.1%) and high (72.9%) school students	Cross-sectional study				
Zhao et al. [90] China	N = 513 (M = 38.21%)	14–18 (M = 16.01, SD = 1.83)		Cross-sectional study				
Han et al. [52] China	N = 1.111 (54.91%)	-	Primary (28.8%), middle (43.74%) and high school (27.45%) students	Cross-sectional study	86.6%	1.89%	8.19%	3.24%
Shin and Choi [92] South Korea	N = 4.958 (51.8)		Elementary (35.1%), middle (33.2%) and high (31.8%) school students	Longitudinal study	-	9.5%	19.7%	6.4%
Vaillancourt et al. [101] Canada	N = 2.683 (50.0%)	M_age_ = 13.05	Primary and secondary school students	Retrospective randomized study	-	2.3%	11.5%	-
Vejmelka and Matković [98] Croatia	N = 494 (42.7%)	12–18 years (M = 14.97)	Middle and high school students	Cross-sectional study	73.08%	5.87%	12.75%	8.3%

Notes: NI = Not involved, CB = Cyberbullying; CV = Cybervictimization; CB/CV = Cyberbullying/victimization.

## Data Availability

The data presented in this study are available on request from the corresponding author.

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
