# Peer review of "Has the COVID-19 Pandemic Affected Cyberbullying and Cybervictimization Prevalence among Children and Adolescents? A Systematic Review"

_ijerph, 2023, doi:10.3390/ijerph20105825_

Round 1

Reviewer 1 Report

This appears to be a worthwhile review, but it is a pity that the findings are rather inconclusive.

Line 31 – ‘Despite’ better than ‘Although’

Line 57  ‘exacerbate’

Around lines 68-76, it may be worth pointing out here the substantial overlap between cyber victimization and cyber bullying – very well established.

Line 84  delete ‘s’ before ‘lives’

Line 105 – ’15 European countries’ -but the title of reference [78] states 11 European countries ?

The remainder of the article is satisfactory.  However I was surprised that there were only 15 studies found.  It is a pity that studies on ‘traditional’ bullying were not included, as a comparison.

Given the great differences among the 15 studies (e.g. lines 331-333), it feels like a review as presented in this article (Tables 1 and 2) might be ‘overkill’.  A narrative review might be more appropriate in results, pointing out methodological differences in discussion. This is perhaps more an editorial decision.

Author Response

Q1: This appears to be a worthwhile review, but it is a pity that the findings are rather inconclusive.

R: We want to thank the reviewer for this recognition and their valuable comments.

Q2: Line 31 – ‘Despite’ better than ‘Although’

Line 57  ‘exacerbate’

R: We thank the reviewer for this suggestion. We changed accordingly.

Q3: Around lines 68-76, it may be worth pointing out here the substantial overlap between cyber victimization and cyber bullying – very well established.

R: We thank the reviewer for this suggestion. Accordingly, we have pointed out the significant overlap between cyberbullying and cybervictimization (i.e., lines 87-90).

Q4: Line 84  delete ‘s’ before ‘lives’

R: We thank the reviewer for this suggestion. We changed accordingly.

Q5: Line 105 – ’15 European countries’ -but the title of reference [78] states 11 European countries?

R: We apologize for this typo. We corrected accordingly (i.e., lines 119-121).

Q6: The remainder of the article is satisfactory.  However I was surprised that there were only 15 studies found.  It is a pity that studies on ‘traditional’ bullying were not included, as a comparison. Given the great differences among the 15 studies (e.g. lines 331-333), it feels like a review as presented in this article (Tables 1 and 2) might be ‘overkill’. A narrative review might be more appropriate in results, pointing out methodological differences in discussion. This is perhaps more an editorial decision.

R: We thank the reviewer for this suggestion. In line with several previous studies on cyberbullying, we consider cyberbullying as the electronic extension of school bullying. However, considering the aim of our systematic review, which is to assess the unique contribution that the Covid-19 pandemic and the consequent lockdown measures could have had on cyberbullying and cybervictimization prevalence rates, school bullying was not included in our research string. Furthermore, the Covid-19 pandemic, even if in different periods across countries, has limited, if not prevented, minors and youth’s face-to-face relationships, thus resulting also in the impossibility of being bullied and/or victimized (Repo et al., 2020; Armitage, 2021) (i.e., lines 130-142).

Reviewer 2 Report

The article"The prevalence of cyberbullying and cybervictimization during COVID-19 pandemic: A systematic review" is methodologically adequate.However, the title is misleading. The authors says at the conclusions that "only few studies, and mainly conducted in Asian countries, have investigated the prevalence of CB/CV during the pandemic". Extrapolating conclusions from Asian studies can be problematic.

Author Response

Q1: The article "The prevalence of cyberbullying and cybervictimization during COVID-19 pandemic: A systematic review" is methodologically adequate. However, the title is misleading.

R: Thank you very much for your feedback. We appreciate your consideration and time in reviewing the manuscript. Accordingly, we changed our manuscript title to “Has the COVID-19 pandemic affected cyberbullying and cybervictimization prevalence among minors? A systematic review”.

Q2: The authors says at the conclusions that "only few studies, and mainly conducted in Asian countries, have investigated the prevalence of CB/CV during the pandemic". Extrapolating conclusions from Asian studies can be problematic.

R: We thank the reviewer one more time also for underlining the methodological appropriateness of our manuscript. The studies surveyed in our systematic review were assessed according to the PRISMA checklist, resulting in 10 Asian studies on a total of 16. Considering the contrasting results that emerged, instead of drawing conclusions concerning the global prevalence of CB and CV during the COVID-19 pandemic, we tried to indicate possible future implications for research. We addressed this issue on lines 521-526.

Reviewer 3 Report

Dear Authors

I do believe you made a good effort to gather the pieces that may explain cyberbullying during COVID 19 pandemic, however, this paper need a lot of maduration in order to be suitable for publication. The next reasons are my main concerns:

1. This manuscript must be revisted and edited by an Native. It is hard to me following the ideas that authors want to deliver.

2. I also noticed you said "A systematic review was conducted in December 2022", and then, you said works from 2023 were also included. 

3. You found nothing relevant to the field. Therefore, I do not know how your work can contribute to the field.

Author Response

Q1: I do believe you made a good effort to gather the pieces that may explain cyberbullying during COVID 19 pandemic, however, this paper need a lot of maduration in order to be suitable for publication. The next reasons are my main concerns:

R: Thank you very much for your feedback. We appreciate your consideration and time in reviewing the manuscript. Below are the replies:

  1. This manuscript must be revisted and edited by an Native. It is hard to me following the ideas that authors want to deliver.

R1: The manuscript has been revised and edited.

  1. I also noticed you said "A systematic review was conducted in December 2022", and then, you said works from 2023 were also included. 

R2: As also reported in the manuscript, our systematic search was carried out in December 2022. In order to include any in press publication, we searched for the time period 2020-2023 (i.e., lines 170-174). However, no results in press were found or included in our systematic review.  

Q3: You found nothing relevant to the field. Therefore, I do not know how your work can contribute to the field.

R3: We thank the reviewer one more time.

Reviewer 4 Report

The prevalence of cyberbullying and cybervictimization during COVID-19 pandemic: A systematic review

Thank you for this very interesting paper. Please see comments below.

Introduction

It is clear that a thorough literature search was undertaken to underpin this SR. The quoted statistics and prevalence reported here warrant an exploration of this topic, but it would be interesting to compare this prevalence with traditional bullying. For example, in Australia, Jadambaa et al. (2019) found that cyberbullying victimisation and perpetration were less common than traditional bullying while in the Nordic context, Arnarsson et al (2020) found that boys were more likely to engage in traditional bullying while girls were more involved in cyberbullying.

In terms of the negative effects of CB, how do these compare with the effects of traditional bullying? Has the pandemic changed these stats at all?

Jadambaa, A., Thomas, H.J., Scott, J.G., Graves, N., Brain, D. and Pacella, R., 2019. Prevalence of traditional bullying and cyberbullying among children and adolescents in Australia: A systematic review and meta-analysis. Australian & New Zealand Journal of Psychiatry53(9), pp.878-888.

Arnarsson, A., Nygren, J., Nyholm, M., Torsheim, T., Augustine, L., Bjereld, Y., Markkanen, I., Schnohr, C.W., Rasmussen, M., Nielsen, L. and Bendtsen, P., 2020. Cyberbullying and traditional bullying among Nordic adolescents and their impact on life satisfaction. Scandinavian journal of public health48(5), pp.502-510.

Results

Interesting that all papers are quantitative papers. Although this probably reflects the exploration of prevalence rates during the pandemic, were qualitative papers excluded from the search and if so, why? If not, can you comment as to why none were picked up in the search?

Although drawing conclusions regarding age was not possible, can any conclusions/inferences to CB/CV be drawn in relation to the gender of those who participated?

Table 1. Definitions and measures of CB/CV.

Each of the 15 studies adopted a different definition of CB/CV.  This could have an impact on the presented results as research shows that children and adults often differ in how bullying is defined. For example, Vaillancourt et al. (2008) found that children were often spontaneous in their definitions and usually omitted elements of repetition, power imbalance and intent. They argue that children should be provided with a bullying definition so similarities and comparisons can be drawn. However, Huang and Cornell (2015) found that young people often use their own perceptions of bullying when answering self-report questionnaires and are not swayed by a provided definition. It would be beneficial to the article to include a brief discussion on this point.  

Vaillancourt, T., McDougall, P., Hymel, S., Krygsman, A., Miller, J., Stiver,K., et al. 2008. Bullying: are researchers and children/youth talking about the same thing? Int. J. Behav. Dev. 32, 486–495.

Huang, F. L., and Cornell, D. G., 2015. The impact of definition and question order on the prevalence of bullying victimization using student self-reports. Psychol.Assess. 27:1484.

In the recommendations provided in the discussion, can you say anything about the responsibilities afforded to social media sites and how they can help to reduce this problem?

Author Response

Thank you for this very interesting paper. Please see comments below.

R: We want to thank the reviewer for this feedback. We appreciate their consideration and time in reviewing the manuscript.

Introduction

Q1: It is clear that a thorough literature search was undertaken to underpin this SR. The quoted statistics and prevalence reported here warrant an exploration of this topic, but it would be interesting to compare this prevalence with traditional bullying. For example, in Australia, Jadambaa et al. (2019) found that cyberbullying victimisation and perpetration were less common than traditional bullying while in the Nordic context, Arnarsson et al (2020) found that boys were more likely to engage in traditional bullying while girls were more involved in cyberbullying.

In terms of the negative effects of CB, how do these compare with the effects of traditional bullying? Has the pandemic changed these stats at all?

Jadambaa, A., Thomas, H.J., Scott, J.G., Graves, N., Brain, D. and Pacella, R., 2019. Prevalence of traditional bullying and cyberbullying among children and adolescents in Australia: A systematic review and meta-analysis. Australian & New Zealand Journal of Psychiatry, 53(9), pp.878-888.

Arnarsson, A., Nygren, J., Nyholm, M., Torsheim, T., Augustine, L., Bjereld, Y., Markkanen, I., Schnohr, C.W., Rasmussen, M., Nielsen, L. and Bendtsen, P., 2020. Cyberbullying and traditional bullying among Nordic adolescents and their impact on life satisfaction. Scandinavian journal of public health, 48(5), pp.502-510.

R: We thank the reviewer for this suggestion. We quoted Amarsson et al. (2020)’s study among those reporting the negative consequences of cybervictimization (i.e., line 71). Despite the significant overlap between school bullying and cyberbullying, COVID-19 and the restrictive lockdown measures had effectively limited children and adolescents' face-to-face interactions, potentially creating the opportunity for increasing their involvement in cyberbullying rather than in school bullying and reversing school bullying trends, whose prevalence rates before the pandemic were much higher than cyberbullying (Jadambaa et al., 2020; Wolke et al., 2017). According to this rationale, our systematic review specifically focuses on cyberbullying and cybervictimization trends during the COVID-19 pandemic (i.e., lines 132 – 146).

Results

Q1: Interesting that all papers are quantitative papers. Although this probably reflects the exploration of prevalence rates during the pandemic, were qualitative papers excluded from the search and if so, why? If not, can you comment as to why none were picked up in the search?

R: Qualitative papers were not excluded from the search. These results, also in line with the reviewer’s suggestion, should be due to the search string used for our systematic review (cyberbully* OR cybervictim*) AND (prevalence OR diffusion) AND (minor* OR adolescent* OR teenager*) AND (“COVID-19” OR “sars-cov-2” OR pandemic OR lock-down) (i.e., lines 209-210).

Q2: Although drawing conclusions regarding age was not possible, can any conclusions/inferences to CB/CV be drawn in relation to the gender of those who participated?

R: We thank the reviewer for this suggestion. Accordingly, we added a paragraph concerning gender differences and involvement in CB and CV  Consistent with previous studies, our systematic review highlights the existence of contrasting results, stressing the importance of taking into account such differences when developing CB/CV measurement questionnaires (Chun et al., 2020). We addressed this issue both in the results (i.e., lines 418-434) and in the discussion (i.e., lines 506-512) sections.

Q3: Table 1. Definitions and measures of CB/CV. Each of the 15 studies adopted a different definition of CB/CV.  This could have an impact on the presented results as research shows that children and adults often differ in how bullying is defined. For example, Vaillancourt et al. (2008) found that children were often spontaneous in their definitions and usually omitted elements of repetition, power imbalance and intent. They argue that children should be provided with a bullying definition so similarities and comparisons can be drawn. However, Huang and Cornell (2015) found that young people often use their own perceptions of bullying when answering self-report questionnaires and are not swayed by a provided definition. It would be beneficial to the article to include a brief discussion on this point.  

Vaillancourt, T., McDougall, P., Hymel, S., Krygsman, A., Miller, J., Stiver,K., et al. 2008. Bullying: are researchers and children/youth talking about the same thing? Int. J. Behav. Dev. 32, 486–495.

Huang, F. L., and Cornell, D. G., 2015. The impact of definition and question order on the prevalence of bullying victimization using student self-reports. Psychol.Assess. 27:1484.

R: We thank the reviewer. We tried to do our best in order to deepen such aspects in the results (i.e., lines 307-332) and in the discussion (i.e., lines 498-506) sections.

Q4: In the recommendations provided in the discussion, can you say anything about the responsibilities afforded to social media sites and how they can help to reduce this problem?

R: We thank the reviewer for this suggestion. Although some social media platforms have set policies to report cyberbullying behaviors and tried to restrict children's use of such platforms, there are no transversal and equally shared rules of conduct among the plethora of existing social media platforms. In this sense, we believe that to protect children and adolescents' physical and mental health, it would be appropriate to include specific curricula on digital safety in cyberbullying prevention programs. Accordingly, we discussed this issue (i.e., lines 588-602).